# Morphological Effect of Side Chain Length in Sulfonated Poly(arylene ether sulfone)s Polymer Electrolyte Membranes via Molecular Dynamics Simulation

**DOI:** 10.3390/polym14245499

**Published:** 2022-12-15

**Authors:** Xue Li, Hong Zhang, Cheng Lin, Ran Tian, Penglun Zheng, Chenxing Hu

**Affiliations:** 1School of Mechanical Engineering, Beijing Institute of Technology, Beijing 100081, China; 2Civil Aircraft Fire Science and Safety Engineering Key Laboratory of Sichuan Province, Civil Aviation Flight University of China, Guanghan 618307, China

**Keywords:** proton exchange membranes, poly(arylene ether sulfone)s, morphological effect, side chain length, molecular dynamics simulation

## Abstract

With the recognition of the multiple advantages of sulfonated hydrocarbon-based polymers that possess high chemical and mechanical stability with significant low cost, we employed molecular dynamics simulation to explore the morphological effects of side chain length in sulfonated polystyrene grafted poly(arylene ether sulfone)s (SPAES) proton exchange membranes. The calculated diffusion coefficients of hydronium ions (H_3_O^+^) are in range of 0.61–1.15 × 10^−7^ cm^2^/s, smaller than that of water molecules, due to the electrical attraction between the oppositely charged sulfonate group and H_3_O^+^. The investigation into the radial distribution functions suggests that phase segregation in the SPAES membrane is more probable with longer side chains. As the hydration level of the membranes in this study is relatively low (λ = 3), longer side chains correspond to more water molecules in the amorphous cell, which provides better solvent effects for the distribution of sulfonated side chains. The coordination number of water molecules and hydronium ions around the sulfonate group increases from 1.67 to 2.40 and from 2.45 to 5.66, respectively, with the increase in the side chain length. A significant proportion of the hydronium ions appear to be in bridging configurations coordinated by multiple sulfonate groups. The microscopic conformation of the SPAES membrane is basically unaffected by temperature during the evaluated temperature range. Thus, it can be revealed that the side chain length plays a key role in the configuration of the polymer chain and would contribute to the formation of the microphase separation morphology, which profits proton transport in the hydrophilic domains.

## 1. Introduction

Alternative hydrocarbon-based proton exchange membranes (PEMs) have been intensively studied for potential application in polymer electrolyte membrane fuel cells (PEMFCs) in automobile transportation, aiming to overcome the drawbacks of the currently widely used perfluorosulfonic acid polymers [1,2,3,4]. PEM serves as a proton conductor and reactants separator, and is required to possess good thermal and hydrolytic stability, remarkable mechanical properties and sufficient water uptake, while maintaining moderate swelling ratio; it is a key component in PEMFCs [5,6]. Sulfonated aromatic polymers are expected to be potential PEM materials due to their low cost, and high chemical and mechanical stability. A range of sulfonated aromatic hydrocarbon-type polymers, including poly(phenylene) [7], poly(ether ether ketone)s (PEEK) [8,9], poly(ether ether sulfone)s (PEES) [10], poly(arylene ether)s (PAE) [11,12] and polyimides (PI) [13], have been investigated as potential PEMs. Among these polymers, sulfonated poly(arylene ether sulfone)s (SPAES) is widely considered a promising alternative material for its outstanding mechanical, thermal and chemical stability. Moreover, the low hydrogen permeability and low-cost nature make it a competitive substitute as a proton exchange membrane [14,15].

SPAES membranes, in pure, composite and blend forms have been investigated and proven to be promising candidates as PEMs. To date, studies of high-proton-conducting SPAES have shown that a promising way to enhance PEM performance and achieve hydration balance is to induce phase-separated morphology with hydrophilic and hydrophobic matrices [16,17,18]. Ding [19], Holdcraft [20] and Guiver [21] investigated the structure–morphology–property relationships and came to the same conclusion that chemical structure and polymer morphology have a strong influence upon water sorption and proton transport properties. Kim [22] reported that SPAES membranes, cross-linked by dihydroxy perfluoropolyether (PFPE), exhibit desirable physicochemical stability and comparable proton conductivity. An effective phase-separated morphology was obtained between the hydrocarbon and perfluorinated moieties, forming well-connected networks; the resulting membrane exhibited notable durability against rigorous operating conditions. Sharma [23] reported a series of acid-base hybrid membranes containing SPAES via the encapsulation of polyethyleneimine(PEI) as a branched pendent. The ionic interactions led to enhanced ionic cluster size and improved proton conductivity, as well as better chemical stability. Lee [24] synthesized SPAES block polymers with aliphatic chains possessing different segment lengths of hydrophilic and hydrophobic oligomers, and the membrane exhibited reasonable dimensional and oxidative stability. Their research provided relevant information, pertinent to designing new block copolymer membranes with desirable physiological, morphological and structural properties. Furthermore, Yuan [25] prepared PEMs based on SPAES with multiple alkylsulfonated side-chains and block copolymer structures; the resulting membrane exhibited high conductivity of 234–319 mS/cm at 80 ℃ and a maximum power density of 1730–2140 mW/cm^2^, higher than that of a Nafion^®^ 212 membrane (117 mS/cm and 1350 mW/cm^2^) tested in the same condition. The enhanced performances are attributed to multiple sulfonated side-chains integrating both rigid aromatic and flexible aliphatic units, and the conformation of microphase separation morphology. The effects of doping inorganic nanomaterials on proton conductivity, power density, fuel crossover, thermal and chemical stability, and durability are reviewed comprehensively by Lade and coworkers [12].

As the functional feature size of PEM is forced on nanoscale, it is important to analyze the membrane structure from a microscopic perspective to give an insightful understanding of the mechanism and to forecast macro-performance. Understanding the effect of microscopic parameters on proton conductivity will help to design and develop PEMs that fit applications at different temperatures and humidity conditions more efficiently. Molecular dynamics (MD) simulation is a powerful tool that provides information on the structure–property relationship at an atomistic level and is mostly employed to perfluorosulfonic acid (PFSA) membranes [26,27]. Sengupta [28] performed a classical molecular dynamics simulation to study a novel multi acid side chain perfluoroimide acid polyelectrolyte membrane and found that the radius of gyration of this system was minimally influenced by hydration and temperature. Meanwhile, the large continuous water cluster was observed at high hydration levels, which would enhance the vehicular diffusion rates. Paddison and coworkers [29,30,31,32,33] have systematically studied the short-side-chain PFSA membranes using MD simulations, focusing on the effects of side chain flexibility and connectivity on hydration and proton transfer. The results indicate that short-side-chains are associated with greater proportions of free hydronium ions and exhibit higher ionic conductivity. Savage [34] performed reactive MD simulations to elucidate the proton transport mechanism in commercial PFSA membranes at different hydration levels and came to the conclusion that proton swapping between sulfonate groups is the primary transport mechanism. Although both vehicular and hopping mechanisms were considered, no evidence has been found in the excess proton self-diffusion constants and a more complete picture of the actual morphology is needed to obtain a better overall agreement [35,36]. Kim [37] used MD simulations to investigate the ion transport properties of modified PEEK polymers with both sulfonic acid and ammonium moiety, and the results are consistent with experimental observations.

In contrast to the extensive experimental studies carried out regarding the morphological and transport properties of SPAES membranes, simulation investigations of these membranes have not yet been sufficiently addressed in previous works. Despite Fatemeh and coworkers’ endeavors [38] to investigate the structural and dynamic properties of SPAES with different level of carboxylation using MD simulation, little work has been carried out to get a theoretical insight into the morphological and transport properties. In this article, we used MD simulations to investigate the morphological effects of the sulfonated side chains on microstructure and proton conductivity of SPAES, which was synthesized and analyzed in our previous work [39]. Molecular models of hydrated SPAES polymer chains with the same hydration number and degree of sulfonation but different side-chain-lengths were constructed, and the molecular diffusive behavior of water molecules (H_2_O) and hydronium ions (H_3_O^+^) inside the SPAES membranes was discussed. 

## 2. Simulation Methodology

Molecular dynamics simulations were performed using the Materials Studio 2017 software (BIOVIA Inc. Paris, France). Amorphous cells containing SPAES polymer chains with different side chain lengths and the corresponding number of hydronium ions and water molecules were constructed to analyze the polymer morphology and molecular transport of the SPAES membranes.

### 2.1. Atomistic Models and Amorphous Cells Construction

The factors primarily affecting the proton conduction is the structural length of the SPAES side chain and resulting sulfonic acid group distributions. As shown in Figure 1, SPAES polymer chains partially grafted with sulfonated polystyrene side chains of different lengths were investigated. The polymer parameters in this work were set based on our previous experimental data at hydration level of λ = 3 and different side chain lengths. The side chains consist of 2 polystyrene repeat units, 6 polystyrene repeat units and 10 polystyrene repeat units; they were set to be at a 50% degree of sulfonation and were constructed as shown in Figure 2b, which later would be used to build homopolymer chains. Each kind of these chains were then built into amorphous cells with the corresponding number of water molecules and hydronium ions to maintain the desired hydration level and electric neutrality of the system. All MD simulations contained 10 sulfonated polystyrene grafted poly(arylene ether sulfone)s copolymer chains, with n = 2, m = 3, x = 2,6 and 10 at 50% degree of sulfonation, as shown in Figure 1. The end of each chain was terminated by hydrogen. In addition, all the sulfonic acid groups were assumed to be ionized. The equivalent weight (EW) of the SPAES membranes in this study was 806.

### 2.2. Simulation Methods

All of the constructed models, including the molecules, ions and polymer chains were pre-optimized by minimizing the total energy with a Smart algorithm and fixing the convergence criteria to 2 × 10^−5^ kcal/mol, 0.001 kcal/mol/Å and 1.0 × 10^−5^ Å, for energy, force and displacement, respectively. An atom based summation method was employed to describe the non-bond interactions with a cutoff distance of 18.5 Å. The charges of each atom were assigned by the forcefield according to the specific chemical environment.

Configurations of all initial amorphous cells were primarily optimized by a Smart algorithm and the same parameters as the optimization of the components were pre-constructed. All MD simulations were based on the DREIDING force field, which effectively predicts the structures and dynamics of various organic, inorganic and biological molecules, featuring the single force constants for each bond, angle, inversion and torsional barrier. Due to the high rigidity of the SPAES backbone, the initial density of the system was set to 0.1 g/cm^3^, which would later go through geometric optimization and an annealing process to achieve higher packing density of 1.1 g/cm^3^ by NPT ensemble; this was to minimize the energy of the system effectively, where N, P, T represents the number of atoms, the pressure and the temperature of the system, respectively. The optimized configuration of the amorphous cells was annealed from 300 K to 550 K until equilibration and the desired density at a ramp of 50 ps, with geometry optimization performed after each cycle. The Nosé–Hoover–Langevin (NHL) algorithm, which contains the Langevin friction and noise term, was used for the thermostat to control the temperature; meanwhile, the Berendsen method was employed, as a barostat changes the coordinates of the particles and the size of the unit cell by employing a re-scaling factor. The polymer chains spread uniformly and interacted with each other during the annealing process, forming clusters with gyration by attractive interaction between hydrophilic domains of the polymer. 

A detailed composition of equilibrated amorphous cells is outlined in Table 1 and snapshots of them are shown in Figure 2a. Dynamics simulation was initiated using the modeling process described above for 500 ps at 300 K, during which the NVT ensemble was applied to identify ionic molecular behavior, where N, V, T represents the number of atoms, volume and temperature of the system, respectively. Although the annealing process led to energy stabilization, the initial 100 ps data were excluded with the concern that unexpected molecular behavior may occur, caused by changing ensembles. For statistical treatment, over three independent models were simulated for each case.

### 2.3. Model Analysis and Property Calculations

As the proton transport by Grotthuss mechanism is not considered in this simulation work, to evaluate molecular conductivity during vehicular conduction of the simulated polymer, it is necessary to define the diffusion coefficient (D), which plays a key role in terms of proton conductivity in MD simulations, as defined by: (1)$$D=\frac{1}{6N}\underset{t\to \infty }{\lim}\frac{d}{dt}\overset{N}{\underset{i=1}{\sum }}\left[{\left(R_{i}\left(t\right)-R_{i}\left(0\right)\right)}^{2}\right]$$
where *N* is the total number of particles, $R_{i}\left(t\right)$ and $R_{i}\left(0\right)$ represent the position vectors of atom *R* at moment *t*, and the beginning of the dynamic calculations, respectively. Mean square displacement (MSD)≡$\;\langle {\left[R_{i}\left(t\right)-R_{i}\left(0\right)\right]}^{2}\rangle$, represents the average distance the atom sets travel during the dynamics calculation, and was employed to compare the mobility of target groups. 

A radial distribution function (RDF) analysis of sulfur and oxygen atoms, in polymer chains and H_3_O^+^, H_2_O molecules is critical to clarify the mechanism of ionic conduction affected by the side chain morphology, resulting from the migration of hydronium among sulfonic acid groups. The RDF, $g_{\left(A-B\right)}\left(r\right)$, indicates the probability density of finding atom B around atom A at an interatomic distance of r over the equilibrium trajectory, as defined by:(2)gA−Br=nB4πr2ΔrNBV
(3)$$n_{B}=4\pi \frac{N_{B}}{V}\int r^{2}g_{\left(A-B\right)}\left(r\right)dr$$
where $n_{B}$ indicates the number of atom *B* located at distance r in a shell of thickness Δr from atom *A*, $N_{B}$ indicates the total number of atom *B* in the system, and V is the total volume of the system.

Generally, the polymer chain would gyrate to ensure energy stabilization and identical atoms would associate due to their natural affinity, such as the sulfur atoms at side chain ends in this study. The radius of gyration (*R_g_*) features the polymer structures, which is defined as the root-mean-square (RMS) distance of the atom sets in the molecule from their common mass center; this is calculated as:(4)$$R_{g}^{2}=\overset{N}{\underset{i}{\sum }}m_{i}s_{i}^{2}/\overset{N}{\underset{i}{\sum }}m_{i}$$
where $m_{i}$ and $s_{i}$ are mass and distance from the mass center of atom i, respectively, and *N* represents the total atom number.

## 3. Results and Discussions

### 3.1. Amorphous Cell Equilibration 

A fully equilibrated amorphous cell is the cornerstone of the simulation validity. In this MD simulation, the amorphous cells were constructed at low density due to the high rigidity of the PAES backbone and to achieve the desired density of 1.1 g/cm^3^ (listed in Table 1), with minor undulation in response to the temperature cycle during the annealing process. Furthermore, parameters including temperature and total potential energy were monitored during dynamic calculations to ensure cell equilibration and achieve the proper distribution of polymer chains and molecules, which exhibit only slight fluctuation, as can be seen in Figure 3. The temperature fluctuates within a relatively tight range of 2.3–2.9 K and the potential energy deviation range is 64.3–92.7 kcal/mol. The temperature and potential energy of simulated systems can be considered as constant with less than 1% deviation. As the energy of the system is a stability criterion, its constancy indicates that the systems reached an equilibrium state and that dynamics calculations based on them are reliable.

### 3.2. Numerical Evaluation of Molecular Diffusion

Although the MD simulation hardly includes the effect of the Grotthuss mechanism on conductivity, the condition of composed systems in this study regards vehicular mechanism to be much more influential on ionic conductance, considering SPAES nanostructure models. MSDs of water molecules and hydronium ions were analyzed to give a statistical description of its dispersal behavior inside the sulfonated SPAES membranes with different side chain lengths at the investigated hydration level.

Figure 4 displays the MSDs of H_2_O and H_3_O^+^ in the simulation systems. The mobility of water molecules and hydronium ions in the SPAES membranes is obtained from the linear regime of MSD. For better comparison, the diffusion coefficients of water molecules and hydronium ions in the sulfonated SPAES simulation system are calculated according to Equation (1) and listed in Table 2. The calculated diffusion coefficients of hydronium ions are in the range of 0.61–1.15 × 10^−7^ cm^2^/s, in accordance with the diffusion ability of proton in sulfonated poly(2,6-dimethyl-1,4-phenylene oxide) (SPPO) and Nafion^®^ membranes; this is similar to the figures calculated and measured by Bahlakeh [40] and Devanathan [41], who reported proton diffusion coefficients ranging from 0.48 × 10^−7^ to 5.27 × 10^−7^ cm^2^/s for hydration numbers of 3–13.5. It is obvious that the diffusion coefficients of H_2_O are higher than that of H_3_O^+^ in all the calculated circumstances. The main source for this discrepancy lies in the contribution of proton transport via hopping mechanism, which was not considered in this MD simulation. Furthermore, the electrical neutrality of H_2_O allows migration with higher freedom, resulting in a higher diffusion coefficient than H_3_O^+^ under the same conditions. The transport of positively charged H_3_O^+^ would be partly orientational owing to the electrostatic interaction with negatively charged sulfonic acid group, affecting the motion trajectory and velocity. A higher diffusion coefficient for both H_2_O and H_3_O^+^ are desirable to better maintain the hydration status of the membrane and higher proton conductivity. The MSD results reveal that longer side chain lengths in the SPAES membrane are preferred for better membrane performance, regarding microscale mass transfer.

### 3.3. Morphological Assessment

The RDFs emphasize membrane nano-structural variation with changing side chain length. RDFs of sulfur–sulfur atom pairs of the SO_3_^−^ groups for varied side chain lengths at hydration levels λ = 3 and 300 K are illustrated in Figure 5, which can give some brief information regarding the interaction between sulfonate groups. The RDFs of sulfur–sulfur atom pairs reveal similar trends, with the first peak occurring at 4.65 Å for all of the three simulated polymers, with the highest peak density for SPAES-2 and displaying similarly for the other two. The second peaks occur at 6.65 Å, 6.37 Å and 5.89 Å for SPAES-2, SPAES-6 and SPAES-10, respectively. The intensity of the first peak shows a downtrend with the increasing side chain length, indicating a decreasing probability of finding a sulfur–sulfur atom pair in two SO_3_^−^ groups in the polymer chain. Furthermore, the sulfur–sulfur atom pair separation decreases slightly with the increasing side chain length, which suggests that phase segregation in the SPAES membrane is more probable with longer side chains, attributed to the interlaced configuration of the polymer chain and a more even distribution of the side chains inside the amorphous cells.

Figure 6 presents the RDFs of sulfur in SO_3_^−^ groups and oxygen in H_2_O, as well as oxygen in H_3_O^+^ atom pair, directly related to the interactions between the sulfonate groups, water molecules, and hydronium ions. The dominant peak in the RDFs of S-O(H_2_O) is observed in the range of 3.43–3.65 Å, with a small bump at 5.19 Å for SPAES-2, as illustrated in Figure 6a. The peak height of the sulfonate–water interaction decreased with side chain length, which indicates that H_2_O molecules are more loosely bound to SO_3_^−^ as the side chain length decreases. The magnitude of the main peak reported is higher than reported for the Nafion membrane with a similar hydration level (λ = 5) [42], as they found a broader distribution of S-O(H_2_O) RDFs at the range of 3.2–4.8 Å; this indicates larger water domains in Nafion membrane. The reason might lie in the difference in side chain lengths and hydration levels of the two calculated systems. The increased hydration level would allow freer chain and water movement, leading to weaker interactions between sulfonic acid groups and water molecules, and a more even sulfonic acid group distribution in the same spatial conformation.

S-O(H_3_O^+^) RDFs represent the sulfonate–hydronium ion interaction, as shown in Figure 6b. It is laconic that the interactions of the positively charged hydronium ions and the negatively charged sulfonic acid groups are comparable for SPAES-2, SPAES-6 and SPAES-10 at the simulated hydration level, as the peak of S-O(H_3_O^+^) RDFs are consistent at 3.67 Å, except that the peak magnitude of SPAES-2 is larger than others. The intensity of peak decreased as the side chain length increased, because the simulated hydration level is relatively low and the actual number of water molecules in the amorphous cell increase correspondingly with side chain length. The better distribution of sulfonic side chain enhanced the water molecules distribution around the SO_3_^−^ groups, resulting in weaker interactions between the sulfur–hydronium ions.

The coordination number is defined as the number of molecules around the target molecule in a given radius. To quantitively evaluate the water clusters and hydronium ions bounded around SO_3_^−^ groups, their coordination number was calculated according to Equation (3) and listed in Table 3. The number of H_3_O^+^ coordinating with the SO_3_^−^ group increased from 1.67 to 2.40 with the increasing side chain length. Here, we consider the hydronium ions to be around the sulfonate group if the distance between the sulfur and oxygen atom pair was less than 4.0 Å. As the sulfur–sulfur distance lies in the range of 3.5–7.5, Å according to Figure 5, we can conclude that some of the H_3_O^+^ ions are bounded (have at least one SO_3_^−^ neighbor), of which a significant proportion appear to be involved in bridging configurations coordinated by multiple SO_3_^−^. The average number of water molecules coordinated with SO_3_^−^ group increased from 2.45 to 5.66, and the computed cutoff radius was up 6 Å, according to Figure 6a, including the relative second peak of SPAES-2. As the hydration level in this simulation is 3, the coordination number of H_2_O around sulfur is higher than 3 for SPAEAS-6 and SPAES-10, indicates that some of the water molecules are shared by multiple SO_3_^−^ groups, which reveals the close distribution of SO_3_^−^ groups and well-connected water clusters. 

The RDFs between the oxygen of water molecules and oxygen of hydronium ions, O(H_2_O)-O(H_3_O^+^), and between oxygen atoms in different water molecules, O(H_2_O)-O(H_2_O), are illustrated in Figure 7. The coordination number represents the number of molecules in the first shell of water molecules around a hydronium ion or a water molecule, and is related to the area under the first peak of RDFs. As can be seen in Figure 7a, the most probable distance of oxygen atom pairs in water molecules and hydronium ions is 2.57 Å, with small fluctuation until 7 Å. In Figure 7b, the main peak occurs at 2.71 Å with similar intensity for all the simulated systems, which is inconsistent with our previous research based on fluorinated backbone grafting (2.73 Å) [43]. The oxygen atom pair distance indicates the formation of a hydrogen bond within H_2_O and H_3_O^+^ [44,45];it also reveals a tendency to form water clusters through which hydronium ions can be transferred and possible pathways for the proton transport in the hopping mechanism. Here, we chose 3.5 Å as the first neighbor cutoff distance between H_2_O and H_2_O, as well as H_3_O^+^ for the calculation of the hydration number based on the data acquired from Figure 7; the results are listed in Table 3. The coordination number of water molecules around the hydronium ions increased from 2.77 to 8.09 with the increasing side chain length. Similar regularity can be discovered as the number of water molecules around the H_2_O molecule increase from 1.81 to 7.69 with the side chain length. Thus, it can be found that both H_2_O and H_3_O^+^ are solvated by a greater number of water molecules when the side chain length increases, indicating that larger water clusters formed inside the membrane system, and promoting the proton transport by vehicular mechanism.

In general, polymer chains would gyrate to ensure energy stabilization and atoms associated according to their natural affinity. The radius of gyration, *R_g_*, is a basically applied to feature the polymer configurations and give a sense of the size of the polymer coil, which can be calculated according to Equation (4). Figure 8 presents the radius of gyration of SPAES membranes. *R_g_* of the SPAES-2, SPAES-6 and SPAES-10 are 12.02 ± 0.09 Å, 13.53 ± 0.28 Å and 14.44 ± 0.07 Å, respectively. The radius of gyration increases with side chain length; longer side chains with stronger hydrophilicity result in larger chain expansion than shorter side chains, due to the hydrophobicity of the backbone.

Dynamics simulation was then performed with SPAES-6 using the modeling process described above for 500 ps at 300 K, 325 K and 350 K, respectively, to assess the influence of temperature on polymer morphology. As shown in Figure 9a,b, at all evaluated temperatures, the RDFs of the sulfur–sulfur atom pair and sulfur–hydronium ion pair are basically unchanged with only a slight fluctuation. The radius of gyration and the corresponding polymer conformation of SPAES-6 polymer at different temperatures are illustrated in Figure 10. While the *R_g_* of SPAES-6 at 300 K and 350K exhibits only a slight discrepancy, 13.53 ± 0.28 Å and 13.44 ± 0.08 Å, respectively, the *R_g_* of SPAES-6 at 325 K shows a decrease of approximately 1.2 Å. The variation of *R_g_* is quite small, indicating a less affected morphology of the polymer chains by temperature due to the excellent dimensional stability of the SPAES membrane. Thus, it can be said that the effect of temperature on the microscopic morphology of these polymer chains is insignificant.

## 4. Conclusions

In summary, an attempt has been made in the present work to describe the observation of sulfonated polystyrene grafted on poly(arylene ether sulfone)s membrane by MD simulation. It is believed that the side chain lengths may have an effect on the performance of the SPAES membranes. Thus, we have devised molecular models for SPAES membranes with the same degree of sulfonation and hydration, but with different side chain lengths to understand the microstructures of the membranes and transport behavior of the H_2_O and hydronium ions. The RDFs of different atom pairs were analyzed to investigate the interaction and their contribution to membrane morphology at the macro level. It was observed that, with increased side chain length, phase segregation in the SPAES membrane is more probable, attributed to the interlaced configuration of the polymer chain and a more even distribution of the side chains inside the amorphous cells. The coordination number of H_2_O and H_3_O^+^ around sulfonate groups increase with the side chain length, and so too do the number of water molecules around H_2_O and H_3_O^+^. Increasing the side chain length would benefit the formation of microphase separation morphology in the membrane, as the hydrophilic side chain would gain more freedom to segregate and form hydrophilic domains. The radius of gyration of the polymer is slightly affected by the side chain length. Furthermore, the SPAES membrane exhibited excellent thermal stability in the microphase scale all through the PEMFC working temperature, indicating a promising dimensional stability of the membrane. Finally, this study can conclude that the side chain length plays a key role in the configuration of the polymer chain and would contribute to the formation of the microphase separation morphology, which is favorable for proton conduction.

## Figures and Tables

**Figure 1 polymers-14-05499-f001:**
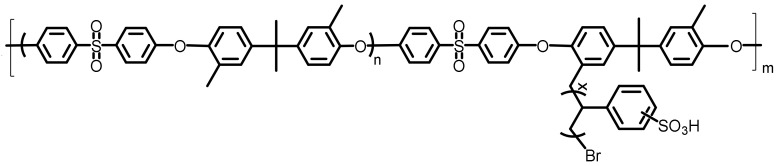
Chemical structure of the simulated SPAES copolymer.

**Figure 2 polymers-14-05499-f002:**
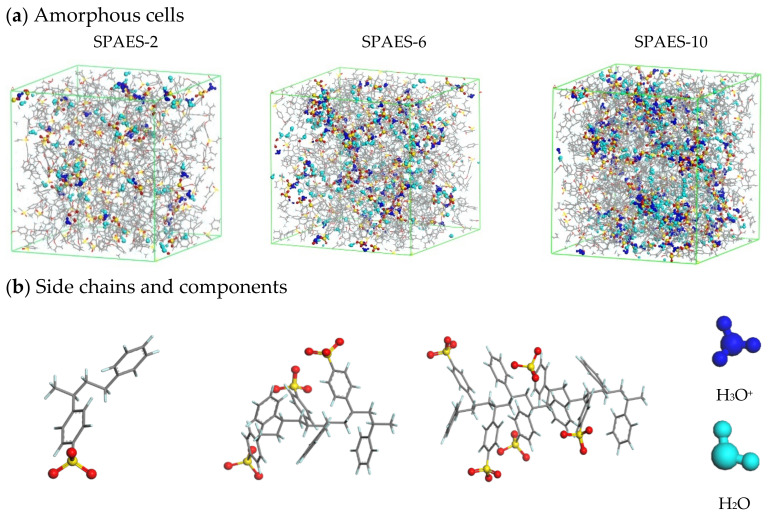
Snapshots of (**a**) the equilibrated amorphous cells and (**b**) components of the polymer side chains, as well as water molecules and hydronium ions. Color scheme in the polymer chains: carbon in gray, hydrogen in light cyan, oxygen in red and sulfur in yellow.

**Figure 3 polymers-14-05499-f003:**
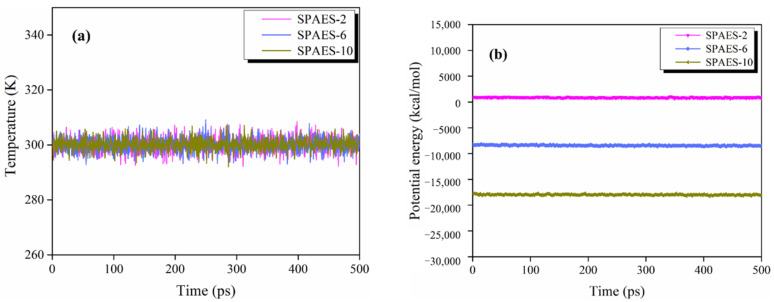
(**a**) Temperature and (**b**) potential energy variation of the systems.

**Figure 4 polymers-14-05499-f004:**
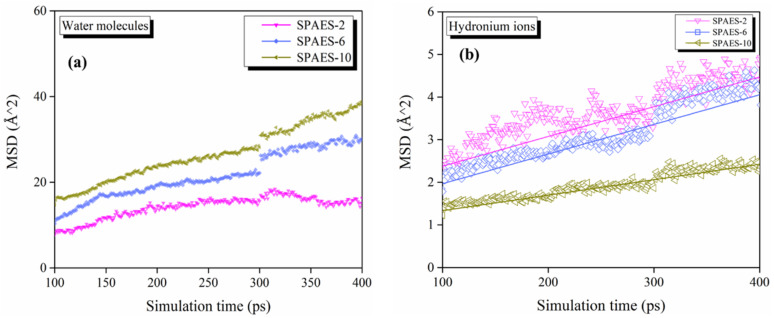
MSDs of (**a**) water molecules and (**b**) hydronium ions inside the SPAES membranes.

**Figure 5 polymers-14-05499-f005:**
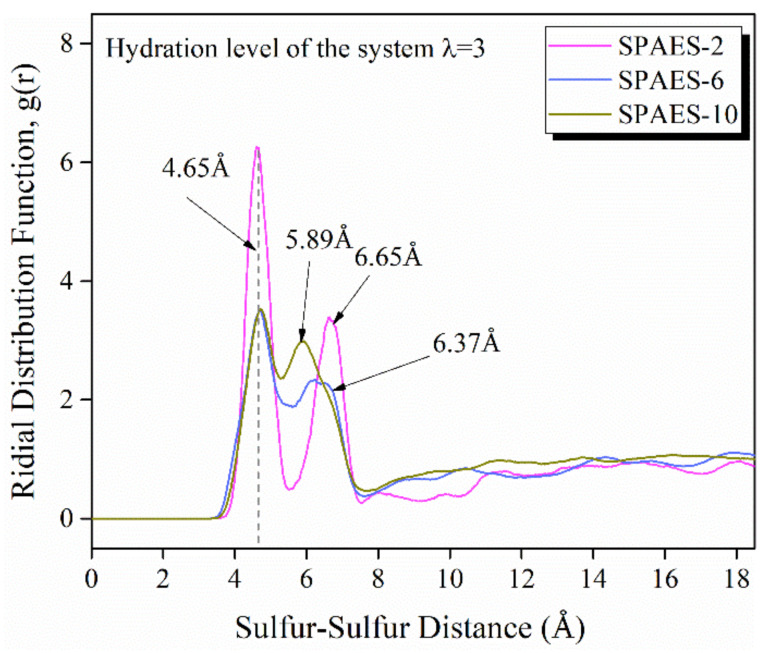
RDFs of sulfur–sulfur for the membrane with various lengths of side chains at hydration level of λ = 3.

**Figure 6 polymers-14-05499-f006:**
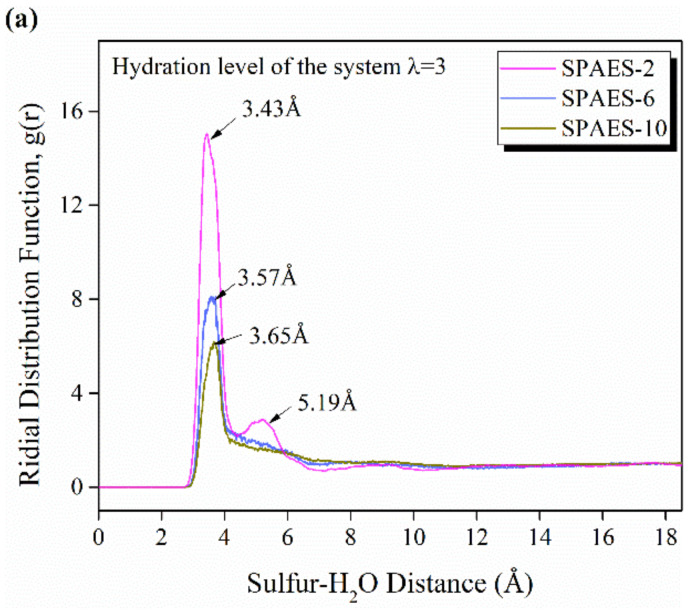
RDFs of (**a**) sulfur–water molecules and (**b**) sulfur–hydronium ions for the membrane with various lengths of side chains at hydration level of λ = 3.

**Figure 7 polymers-14-05499-f007:**
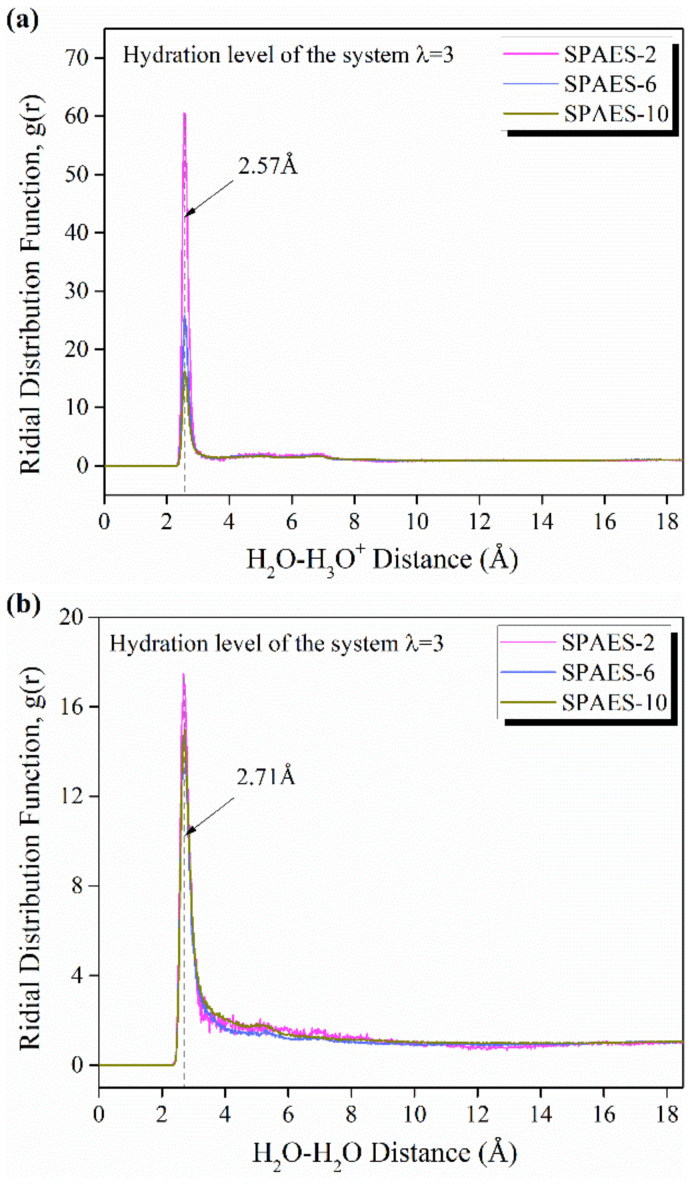
RDFs of water molecules and (**a**) hydronium ions as well as (**b**) water molecules for the membrane with various side chain length of 2, 6 and 10, respectively, at hydration level of λ = 3.

**Figure 8 polymers-14-05499-f008:**
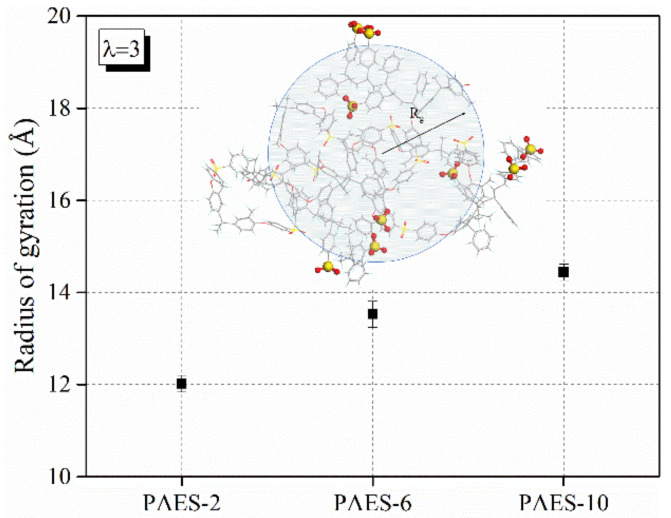
The radius of gyration of the SPAES polymers.

**Figure 9 polymers-14-05499-f009:**
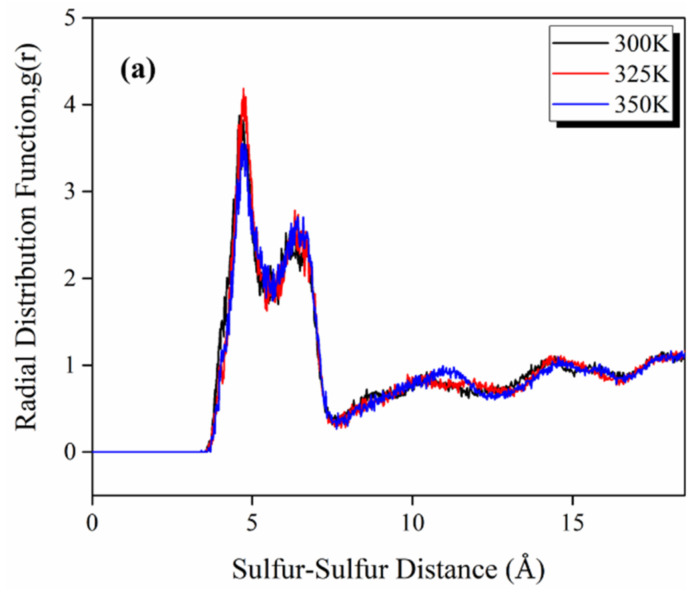
RDFs of (**a**) sulfur–sulfur atom pair and (**b**) sulfur–hydronium ion pair in SPAES-6 polymer at different temperatures.

**Figure 10 polymers-14-05499-f010:**
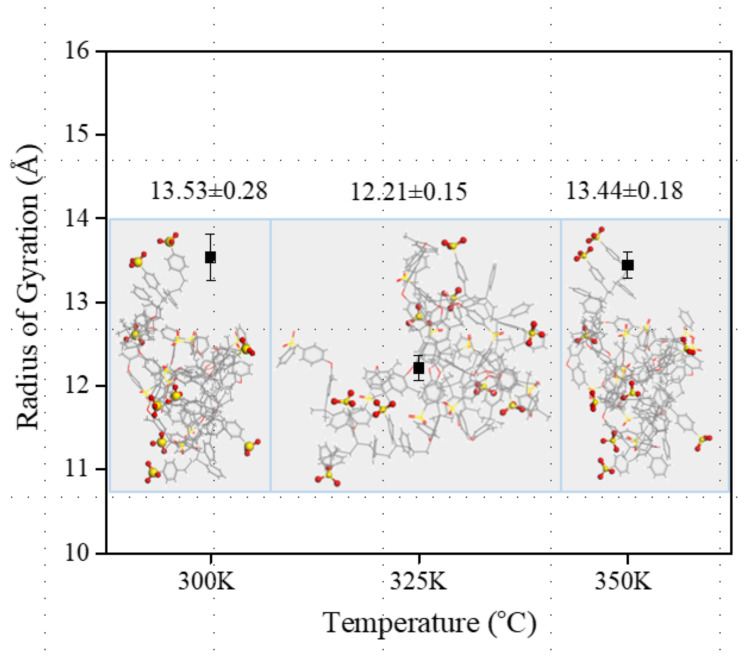
The radius of gyration of the SPAES-6 polymers at different temperatures.

**Table 1 polymers-14-05499-t001:** Composition and parameters of the amorphous cells.

Polymer Cells		SPAES-2	SPAES-6	SPAES-10
Number of Chains		10		
Number of SO_3_^−^		30	90	150
Number of H_3_O^+^		30	90	150
Number of H_2_O		60	180	300
Number of atoms		6680	9380	12,080
Volume (Å^3^)		76,726.60	105,689.08	134,630.27
Temperature (K)	Average	300.01	300.01	299.99
	Std.Dev	2.95	2.56	2.27
Density (g/cm^3^)	Average	1.10	1.10	1.11
	Std.Dev	0.02	0.02	0.02

**Table 2 polymers-14-05499-t002:** Diffusion coefficients of H_2_O and H_3_O^+^ in the hydrated SPAES membranes.

	SPAES-2	SPAES-6	SPAES-10
D_H2O_ (×10^−6^ cm^2^/s)	0.49	1.01	1.21
D_H3O+_ (×10^−7^ cm^2^/s)	1.15	1.16	0.61

**Table 3 polymers-14-05499-t003:** Calculated coordination number at 300 K and 1 atm, λ = 3.

Membranes	SPAES-2	SPAES-6	SPAES-10
O(H_3_O^+^)-S(SO_3_^−^)	1.67	2.12	2.40
O(H_2_O)-S(SO_3_^−^)	2.45	3.51	5.66
O(H_2_O)-O(H_2_O)	1.81	3.42	7.69
O(H_2_O)-O(H_3_O^+^)	2.77	4.62	8.09

## Data Availability

The data presented in this study are available on request from the corresponding author.

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
