# Peer review of "Morphological Effect of Side Chain Length in Sulfonated Poly(arylene ether sulfone)s Polymer Electrolyte Membranes via Molecular Dynamics Simulation"

_polymers, 2022, doi:10.3390/polym14245499_

Round 1

Reviewer 1 Report

The present manuscript is to report on MD results based in sulfonated PAES. The introduction covers a very precise history of hydrocarbon typed polyelectrolytes along simulation. The method is well described. The results and discussions are to be improved.

I see the authors report on results mainly, not discussions. For those who synthesize polymers, it is important to understand which direction needs to be considered. For example, diffusion coefficients and their values are described, but no argument is found telling if a higher or smaller diffusion coefficient is better as PEMs. Should it be close to that of Nafion? How should a reader consider of these results into how to synthesize in light of which parameter? All other results need to be reinforced by such a debate.

Therefore, I recommend the authors to improve their manuscript in particular in Results and Discussions. 

Author Response

We want to thank teh Reviewer for the recognition with our work and consider it to be interesting.  We appreciate the constructive and insightful criticism and advice. As the main concern of the reviewer about this manuscript is the insufficent discussion about the results, we reinforced the explanation on the morphology and dynamic property of the SPAES membrane in the manuscript, coming to an conclude about how the micromorphology vary with side chain length.  We would very appreciate it if you are pleased with the revision and thanks again.

Reviewer 2 Report

Dear Authors

The presented work in your manuscript is interesting for the readers.

With the recognition of the multiple advantages of sulfonated hydrocarbon-based polymers that possessed high chemical and mechanical stability with significantly low cost, we employed molecular dynamics simulation to explore the morphological effects of side chain length in sulfonated polystyrene grafted poly(arylene ether sulfone)s proton exchange membranes. The calculated diffusion coefficient of hydronium ions was in the range of 0.61-1.15 ×10-7 cm2/s, smaller than that of water molecules due to the electrical attraction between the opposite charge of the sulfonate group and H3O+. The radial distribution functions investigation suggests that phase segregation in the sulfonated polystyrene graft poly(arylene ether sulfone)s (SPAES) membrane is more probable with longer side chains. As the hydration level of the membranes in this study is relatively low, longer side chains correspond to more water molecules in the amorphous cell, which would pro-vide better solvent effect for the distribution of sulfonated side chains. The coordination number of water molecules and hydronium ions around sulfur increased from 1.67 to 2.40 and from 2.45 to 5.66 with increasing side chain length, respectively. A significant proportion of the hydronium ions appears to be in bridging configurations coordinated by multiple SO3-. The microscopic confirmation of the SPAES membrane is basically unaffected by temperature during the evaluated temperature range. Thus, it can be revealed that the side chain length plays a key role in the configuration of the polymer chain and would contribute to the formation of the microphase separation morphology.

The main concern in the reading of your manuscript is the difficulty to follow the references which are cited in an improper style. Please cite the references in the journal style.

Also, revise the language throughout the manuscript.

In conclusion, a major revision is required. 

Author Response

We want to thank the Reviewer  for the constructive criticism and suggestion, we addressed all the points raised by the reviewer, as summarized below:

  1.  We have changed the cite style to fit the journal style, some of them are deleted or changed corresponding to the revision of the manuscript. We corrected some typos in the paperwork.
  2. Per the reviewer’s suggestion, werevised the language throughout the manuscript very carefully. 

Round 2

Reviewer 1 Report

I recommend to publish its revised manuscript, as some improvement were made to get readers more attractive than before.

Reviewer 2 Report

Dear Authors

Thanks for taking the comments into consideration during the revision process.

I can recommend the manuscript in its current version for publication.